# Appropriate Lymph Node Dissection Sites for Cancer in the Body and Tail of the Pancreas: A Multicenter Retrospective Study

**DOI:** 10.3390/cancers14184409

**Published:** 2022-09-11

**Authors:** Kimitaka Tanaka, Yasutoshi Kimura, Tsuyoshi Hayashi, Yoshiyasu Ambo, Makoto Yoshida, Kazufumi Umemoto, Takeshi Murakami, Toshimichi Asano, Toru Nakamura, Satoshi Hirano

**Affiliations:** 1Department of Gastroenterological Surgery II, Hokkaido University Faculty of Medicine, Sapporo 060-8648, Japan; 2Department of Surgery, Surgical Oncology and Science, Sapporo Medical University School of Medicine, Sapporo 060-8556, Japan; 3Center for Gastroenterology, Teine-Keijinkai Hospital, Sapporo 006-8555, Japan; 4Department of Surgery, Teine-Keijinkai Hospital, Sapporo 006-8555, Japan; 5Department of Gastroenterological Surgery, Kin-ikyo Chuo Hospital, Sapporo 007-8505, Japan

**Keywords:** pancreatic cancer, distal pancreatectomy, lymph node metastasis, spleen preservation, efficacy index

## Abstract

**Simple Summary:**

Distal pancreatectomy has become the standard surgery for patients with pancreatic body–tail cancers. The optimal area for dissection remains controversial, and there is an emphasis on preserving as much tissue as possible while eradicating the cancer. This multicenter, retrospective study evaluated the frequency and patterns of lymph node metastasis based on tumor site among 235 patients with pancreatic body–tail cancers to determine the optimal lymph node dissection area for distal pancreatectomy. Patients with pancreatic body cancer tumors showed no metastasis to the splenic hilum lymph node. Spleen-preserving pancreatectomy might be feasible for Pb cancer.

**Abstract:**

Distal pancreatectomy (DP) with lymphadenectomy is the standard surgery for pancreatic body–tail cancer. However, the optimal lymph node (LN) dissection area for DP remains controversial. Thus, we evaluated the frequency and patterns of LN metastasis based on the tumor site. In this multicenter retrospective study, we examined 235 patients who underwent DP for pancreatic cancer. Tumor sites were classified as confined to the pancreatic body (Pb) or pancreatic tail (Pt). The efficacy index (EI) was calculated by multiplying the frequency of metastasis to each LN station by the five-year survival rate of patients with metastasis to that station. LN metastasis occurred in 132/235 (56.2%) of the patients. Patients with Pb tumors showed no metastasis to the splenic hilum LN. Distal splenic artery LNs and anterosuperior/posterior common hepatic artery LNs did not benefit from dissection for Pb and Pt tumors, respectively. In multivariate analysis, splenic artery LN metastasis was identified as an independent predictor of poor overall survival in patients with pancreatic body–tail cancer. In conclusion, differences in metastatic LN sites were evident in pancreatic body–tail cancers confined to the Pb or Pt. Spleen-preserving pancreatectomy might be feasible for Pb cancer.

## 1. Introduction

Surgical resection for pancreatic ductal adenocarcinoma (PDAC) provides a chance for long-term survival; however, the survival after curative pancreatectomy is not long enough [1,2]. The treatment for PDAC is shifting away from surgery alone to include multidisciplinary treatments, such as neoadjuvant or adjuvant therapy, to improve prognosis [2,3]. From this perspective, early introduction of adjuvant treatment and improvement of postoperative tolerability are important to enhance the synergistic effect of surgical and nonoperative treatments, such as chemotherapy. In general, major surgery induces tissue damage followed by an extensive immunosuppressive response. Intraoperative blood transfusion further augments the predominately immunosuppressive response seen secondary to tissue damage [4]. Perioperative blood transfusion following pancreatectomy for PDAC is associated with poor short- and long-term outcomes [5]. Immune suppression following surgery has been implicated in increased short-term postoperative complications, especially infectious complications [4]. Therefore, treatment of PDAC may require surgery that does not induce a postoperative immunosuppressive response. Although organ-preserving surgery for PDAC has not yet been reported, it is expected to be a key factor in the surgical treatment of PDAC.

In general, the standard operation for PDAC is thought to be pancreatectomy with lymphadenectomy, which involves dissecting an area based on lymphatic flow from the primary tumor. The recommended extent of lymph node (LN) dissection for distal pancreatectomy (DP) has been defined in the General Rules for the Study of Pancreatic Cancer (7th edition) [6] and the consensus statement by the International Study Group on Pancreatic Surgery (ISGPS) [7]. However, unlike pancreatoduodenectomy, no clinical studies have considered the optimal lymphadenectomy for DP, which remains controversial [8,9,10,11,12].

We have shown the difference in favorable LN metastasis based on tumor site in a single institution [13]. We found that patients with pancreatic body cancer had no metastasis to splenic hilar LNs. In pancreatic body or tail cancers that are eligible for DP, metastasis to LN stations far from the primary tumor, including regional LN stations, is rare. The purpose of the present multicenter study was to evaluate the difference in the frequency of LNs metastasis based on the primary tumor site.

## 2. Materials and Methods

### 2.1. Patients’ Selection

This multicenter retrospective cohort study was conducted by the Hokkaido Pancreatic Cancer Study Group (HOPS). All patients who underwent DP for PDAC between January 2001 and July 2017 at tertiary referral hospitals in Hokkaido Prefecture were enrolled. We excluded patients with intraepithelial cancer (Tis), acinar cell carcinoma, and anaplastic carcinoma. The institutional review boards of Hokkaido University Hospital (018-0317) and each participating hospital approved the study protocol, and the requirement for informed consent was waived. Comprehensive informed consent was obtained from all participants before surgery using the patient information of this study. 

### 2.2. Operation Procedure

DP and D2 dissections were performed as standard procedures. In cases where the tumor was located close to the celiac axis (CA), DP with celiac axis resection (DP-CAR) was performed [14,15]. The indications for DP-CAR were (1) tumors contacting with the common hepatic artery (CHA) and/or the CA, with a cancer-free margin of at least 5–7 mm ligated at its origin from the aorta, and (2) tumors close to the root of the splenic artery within 10 mm that don’t contact with either the CHA or CA [13].

### 2.3. The Definition of Tumor Location

The tumor site was categorized by preoperative multidetector-row computed tomography and/or ultrasonography (US). The distal pancreas is defined as the left border of the superior mesenteric vein (SMV) and portal vein (PV), while a line bisecting the distal pancreas was considered the border between the body and tail of the pancreas [16]. The pancreatic neck is included in the pancreatic body in this study, which is located anterior to the SMV/PV and to the left side of the gastroduodenal artery. The tumor sites were classified as Pb (pancreatic body), Pbt (both the body and tail of the pancreas), or Pt (pancreatic tail) (Figure 1). The pathological stage, number of LNs, and LN dissection criteria were determined according to the General Rules for the Study of Pancreatic Cancer (6th edition) [16]. The number of LNs was defined as follows: LN in the anterosuperior group along the CHA (#8a), LN in the posterior group along the CHA (#8p), LN in the splenic hilum (#10), LN around the proximal splenic artery (#11p), LN around the distal splenic artery (#11d), LN around the proximal superior mesenteric artery (#14p), LN in the para-aorta region (from the lower margin of the left renal vein to the upper margin of the inferior mesenteric artery) (#16b1), and LN along the inferior margin of the pancreas (#18) (Figure 1). Specimens were cut into 5 mm thickness /section on a sagittal plane, mounted on slides, and stained with hematoxylin and eosin. The location of the LN metastasis was confirmed by pathological reports with details of the metastatic sites.

### 2.4. The Efficacy Index Assessment for Each Nodal Station

We analyzed the relationship between the LN metastasis at each station and the prognosis using the efficacy index proposed by Sasako [17]. The efficacy index was a measure of survival benefit calculated by multiplying the frequency of metastasis to the station by the 5-year survival rate and 2-year disease-free survival rate of patients who metastasized to that station.

### 2.5. Multidisciplinary Treatment

Neoadjuvant treatment (NAT) has been administered through clinical studies since 2014, although there are differences in participation criteria among institutions. The use of adjuvant therapy depended on the treatment policy at each institution and on the patient’s postoperative condition. Adjuvant therapies, such as gemcitabine or S-1, were initiated in clinical studies in 2007, and S-1 was established as a standard adjuvant therapy in 2012.

### 2.6. Follow-Up and Recurrent Data

Postoperative follow-up investigations consisted of a physical examination, laboratory studies, and CT imaging at 3- to 4-month intervals for the first two years, at 6-month intervals for 3–5 years, and then at yearly intervals thereafter. Recurrent patterns and survival data were retrospectively collected in December 2019.

### 2.7. Statistical Analysis

Continuous variables are expressed as medians and interquartile ranges. Categorical variables are expressed as numbers and percentages. For comparisons of categorical data, the χ^2^ test or Fisher’s exact test was used. The Mann–Whitney U test was used to compare continuous data. The Wilcoxon test was used to compare the three groups. Overall patient survival (OS) was calculated from the date of surgery to the date of the last follow-up or the date of patient death. Disease-free survival (DFS) was calculated from the date of surgery to the date of the last follow-up or confirmation of recurrence. The Kaplan–Meier method was used to estimate OS and DFS, and survival differences were analyzed using the log-rank test based on the comparison of tumor site and pathological node category. The stepwise selection was performed based on Akaike’s information criterion (AIC), which included factors that were statistically significant in the univariate analysis (*p* < 0.1), and a multivariate analysis was performed using the Cox proportional hazard model. A *p* significance was set at *p* < 0.05. The JMP Pro 16.0 statistical software program (JMP Japan, Tokyo, Japan) was used for all analyses.

## 3. Results

### 3.1. Characteristics of the Entire Population

In total, 235 patients who underwent DP were enrolled. The distribution of patients is shown in Table 1. Fifty-six patients were enrolled from Hokkaido University, 59 from Sapporo Medical University, 101 from Teine-Keijinkai Hospital, and 19 from Kin-ikyo Chuo Hospital. The clinical and pathological characteristics of the patients were compared among the primary tumor sites. LN metastasis occurred in 132/235 (56.2%) patients, with 56 (51.4%), 29 (56.9%), and 47 (62.7%) tumors located in the Pb, Pbt, and Pt, respectively. NAT was administered to 43 (18.3%) patients. The patients who were administered the NAT treatment did not differ significantly between tumor sites.

The operation time was longer in patients with Pb tumors than in those with Pt tumors (*p* = 0.042), but the Pt tumors were larger than the Pb tumors (*p* = 0.040). The rates of pPV and pOO positivity were higher for Pt tumors than they were for Pb tumors (*p* = 0.015 and *p* < 0.001, respectively). The number of harvested LNs was higher in patients with Pb tumors than in those with Pt tumors (*p* = 0.022).

Regarding the characteristics of the patients with Pbt tumors, the proportion of male patients was significantly higher compared with that in the other tumor types (*p* = 0.023). Patients with Pbt tumors underwent DP-CAR more frequently than those with Pb or Pt tumors (*p* < 0.01). Patients with Pbt tumors had a longer operation time, larger blood loss, higher incidence of CD IIIa or more, and longer postoperative hospital stays (*p* < 0.01, *p* < 0.01, *p* = 0.021, and *p* = 0.003, respectively) than did those with other tumor types. Regarding the pathological findings, the Pbt tumor was larger than the other tumor categories (*p* < 0.01). The rate of positive pPV was higher for Pbt tumors than for other tumors (*p* = 0.04).

### 3.2. The Pattern of LN Metastasis According to Tumor Site

Table 2 shows the frequency of LNs metastasis based on the tumor site. LN metastases to #11d, #11p, and #18 were found in >15% of all patients. In addition, >10% of the patients with Pb tumors had #8a/p LN metastasis, while those with Pt tumors had #10 LN metastasis. The efficacy index was calculated using the 5-year OS rate (Table 2) and 2-year DFS rate (Appendix A).

Patients with Pb tumors had no metastases to the #10 LNs. Five-year survivors did not have Pb tumors with metastatic spread to #11d LNs. Regarding the efficacy index assessment, #8a/p, 11p, #14p, and #18 LNs benefited from LN dissection in Pb tumors, whereas in #11d LNs, the 5-year OS index was 0 points and the 2-year DFS index was very small (0.92). Therefore, metastasis to #11d LN considerably worsened the prognosis of patients with Pb tumors.

In Pt tumors, five-year survivors did not have metastatic spread to #8a/p LNs. Regarding the efficacy index assessment, both the 5-year OS index and 2-year DFS index were 0 points in #8a/p LNs. Further, #8a/p LNs did not benefit from LN dissection, even when including regional LNs because of the poor prognosis equivalent to distant metastasis as para-aortic #16b1 node metastasis.

### 3.3. Long-Term Outcomes

In the 235 patients, 18 clinicopathological variables related to OS were analyzed using the Cox proportional hazard model. In the univariate analyses, 10 of the 18 variables were statistically significant (Table 3). For multiple analyses, a stepwise selection was performed based on the AIC. When portal venous system invasion, invasion of other organs, and #11p and #18 LN metastasis were excluded, and eight factors were selected, the AIC value was minimized (1247.96). In a multivariate analysis, the following four factors were identified as independent poor prognostic factors: preoperative CA19-9 values >100 U/mL, tumor size >20 mm, #11d LN metastasis, and no adjuvant treatment (Table 3).

DFS and OS are shown in Figure 2. The median DFS and OS were 725 and 1298 days, respectively. The five-year DFS and OS rates were 30.5% and 32.2%, respectively (Figure 2). Patients with any LNs metastases and those with #11d LNs metastasis had a poorer prognosis (*p* < 0.01, and *p* < 0.01, respectively) than those without (Figure 2). The median OS was 606 days in patients with #11d LNs metastasis.

## 4. Discussion

The frequency of LN metastasis after DP for pancreatic body and tail cancers was summarized in this multicenter study. Our study showed that metastasis to #11 and #18 LNs, which are generally defined as the region of D1 LN dissection, occurred in more than 10% of the patients. In the analysis by tumor localization, no patient with Pb cancer was observed to have metastasis to #10 or #11d LNs, and those with Pt cancer did not have metastasis to #8a/p LNs, demonstrating a zero-efficacy index despite regional LNs. Patients with #11d LNs metastasis had a poor prognosis as an independent poor prognostic factor.

The poor prognostic factors in our study were similar to those of previous reports [1,2,18,19,20]; the preoperative CA19-9 value and pN were previously identified as a biological and pathological poor prognostic factor, respectively. In this study, #11d LN metastasis was an independent poor prognostic factor. Therefore, patients with #11d LN metastasis after DP would require more intense postoperative therapy. Adjuvant therapy was also identified as an independent prognostic factor in this study. However, caution must be used when interpreting these data considering the recent advancement of therapeutic agents after recurrence because many of the patients did not receive adjuvant therapy in the past.

This study was conducted at multiple institutions to validate the results of a previous study performed in a single-center study conducted at Hokkaido University [13]. The major difference between the previous study and the present study is the definition of the boundary between the pancreatic body and tail. The boundary in the previous study was set at the left border of the aorta as defined by the General Rules for the Study of Pancreatic Cancer (7th edition) [6]. On the other hand, in the present study, the boundary was set as a line bisecting the distal pancreas from the left border of the SMV/PV as defined by the General Rules for the Study of Pancreatic Cancer (6th edition) [16]. Although patient-dependent, the boundary of this study is a few centimeters closer to the splenic hilum than to the left border of the aorta. By setting the boundary between the pancreatic body and tail more toward the spleen, we would like to ensure that there is no LN metastasis at the splenic hilum, even when the tumor is located closer to the splenic hilum.

The most frequent sites of LN metastasis were in the D1 dissection area, relatively close to the tumor, as described in the General Rules for the Study of Pancreatic Cancer (6th edition) [16]. Although the recommended extent of dissection is the same for tumors in both the body and tail of the pancreas in the rules, our study demonstrated a large difference in LN metastasis sites between tumors confined to the pancreatic body and those confined to the pancreatic tail. In our analysis using the efficacy index proposed by Sasako [17], we found that #11d LNs for Pb cancer and #8 LNs for Pt cancer had an efficacy index of zero despite being a regional LN. Metastasis to these lymph nodes suggests a poor prognosis, equivalent to distant metastasis, such as para-aortic #16b1 node metastasis. Therefore, more intensive preoperative chemotherapy is required when those LN metastases are suspected radiologically or proved pathologically. In addition, no patients with Pb cancer had LN metastasis to the splenic hilum, which leads us to surmise that the splenic hilum LN (#10) may not need to be dissected. Although our previous study [13] and other studies in Japan, Korea, and France [21,22,23] have reported about the splenic hilum LN after distal pancreatectomy, this is the first multicenter report to describe the detailed frequency of lymph node metastasis with a clearly defined tumor localization.

As a surgical strategy for pancreatic body cancer, spleen-preserving DP (Warshaw procedure) as an organ-preserving surgery is also an option because there is no LN metastasis to the splenic hilum. The merits of spleen-preserving surgery include conservation of splenic function, such as avoidance of the long-term risk of post-splenectomy sepsis related to encapsulated bacteria [24]. The risk of overwhelming post-splenectomy infection (OPSI) is estimated to be 1 per 400–500 patient-years, and that of fatal OPSI is 1 per 800–1000 patient-years [25]. Aiolfi et al. [26] reported that splenectomy for gastric cancer is significantly associated with postoperative infectious complications and overall morbidity. The risk of thromboembolic complications and OPSI, a rare but potentially lethal complication, increases with splenectomy. However, the operation technique to preserve blood flow around the splenic hilum could be difficult due to variations in vessels, inflammation, or obesity.

DP-CAR is often performed in patients with locally advanced pancreatic cancers. Ischemic gastropathy is sometimes a problematic postoperative complication of DP-CAR [27,28]. This ischemic gastropathy has been reported to occur particularly in the posterior wall of the fornix [29]. Left gastric artery preservation or arterial and venous reconstruction is useful for ischemic gastropathy, combining intraoperative ICG to confirm the blood flow after reconstruction; however, issues, such as congestion or early arterial occlusion after reconstruction, remain. One possible cause of ischemic gastropathy is the division of the short gastric artery. It is assumed that the blood flow of the fornix is relatively weak, with only the supply from the capillaries flowing within the stomach. The spleen-preserving DP-CAR (Warshaw procedure) for locally advanced pancreatic body cancer may be a technique that can solve the disadvantages of DP-CAR by avoiding ischemia of the fornix to preserve the connecting tissue, including vessels around the splenic hiatus, such as the short gastric artery and vein and left gastroepiploic artery and vein.

This study has several limitations, including its retrospective nature. It is unclear whether the omission of LN dissection based on tumor localization is oncologically justified, and the feasibility of the spleen-preserving procedure requires further clarification. Thus, as a next step, additional validation needs to be performed in a prospective trial. The laboratory data representative of immune response and nutritional status should be included in the multivariate analysis of long-term outcomes, but the wide variation among centers and time periods made analysis difficult for the multicenter retrospective study. Even though the patient selection differed from our previous study [13], such as the definition of the boundary between the pancreatic body and tail, and the inclusion of NAT patients, 42 patients were in both this and the previous study. The efficacy index assessment could be overestimated in the lymph node station with low rate of metastasis.

## 5. Conclusions

Differences in the metastatic LN sites were evident in pancreatic body-tail cancer when tumors were confined to the left or right side of a line bisecting the distal pancreas from the left border of the SMV/PV. In pancreatic body cancer, no patient had #10 LN metastases, suggesting that spleen-preserving pancreatectomy is feasible for pancreatic body cancer. Finally, #11d LNs metastasis was an independent poor prognostic factor.

## Figures and Tables

**Figure 1 cancers-14-04409-f001:**
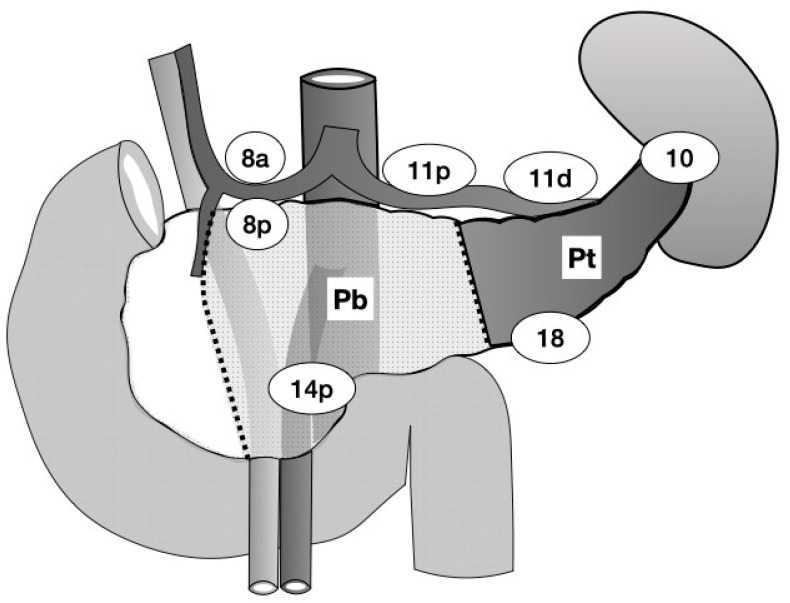
The dotted line indicates the border of each location. The dark area indicates the pancreatic tail (Pt) tumor site. The light-dotted area indicates the site of the pancreatic body (Pb) tumor. Pbt tumors spread to both the Pb and Pt sites. The number in the circle indicates the name of lymph node station according to the Japanese Classification of Gastric Carcinomas.

**Figure 2 cancers-14-04409-f002:**
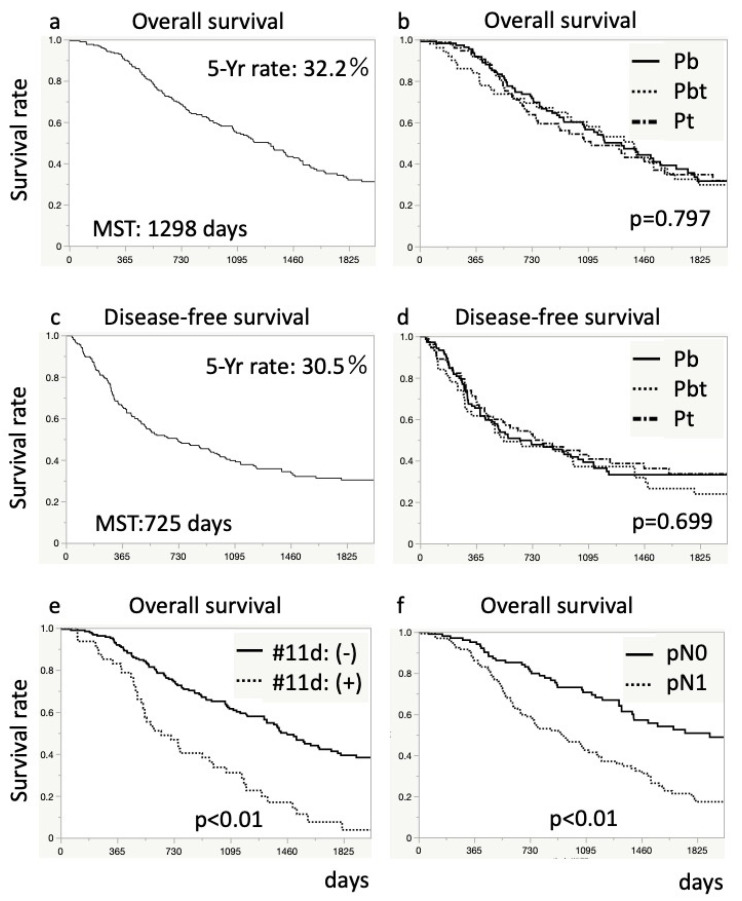
Kaplan–Meier survival curves for patients after distal pancreatectomy. The upper graphs (**a**,**b**) show overall survival rate, and the middle graphs (**c**,**d**) show disease-free survival rate. The graphs (**a**,**c**) present survival rate for all patients. The graphs (**b**,**d**) show a comparison in survival rates based on the tumor site. The lower graphs (**e**,**f**) show an overall comparison of survival rates based on lymph node metastasis: (**e**), pathological node category; and (**f**), #11d lymph node metastasis. Differences in survival were analyzed using log-rank tests.

**Table 1 cancers-14-04409-t001:** Demographic and surgical characteristics of patients who underwent distal pancreatectomy for pancreatic cancer.

	Total Patients *n* = 235	Pb *n* = 109	Pt *n* = 75	*p* Value (Pb vs. Pt)	Pbt *n* = 51	*p* Value (Pbt vs. Pb + Pt)
Gender (Male/Female)	138/97	61/48	40/35	0.725	37/14	**0.023**
Age	70 [63–75] *	72 [63.5–77] *	69 [62–75] *	0.467	69 [63–73] *	0.214
BMI	22.9 [20.6–25.1] *	22.5 [19.7–24.3] *	22.7 [21.0–25.0] *	0.317	23.5 [21.2–25.9] *	0.063
ASA (1/2/3)	39/187/8	23/84/2	9/63/2	0.283	7/40/4	0.129
CA19-9	49.6 [14.3–184.5] *	48.3 [14.3–143.6] *	55 [14.2–160.4] *	0.790	57.2 [13.2–356] *	0.378
Neoadjuvant therapy	43 (18.3%)	26 (23.9%)	11 (14.7%)	0.127	6 (11.8%)	0.173
Hospital				0.770		**<0.001**
HU	56	17	14		25	
SMU	59	31	17		11	
KJ	101	52	39		10	
Kin	19	9	5		5	
Operation procedure				0.124		**<0.001**
DP-CAR	19	8	0		11	
DP (SPDP)	216 (1)	101 (1)	75		40	
Operation time (min)	292 [223.5–377.5] *	290 [233–364] *	268 [204–330] *	**0.042**	367 [276–431] *	**<0.001**
Blood loss (mL)	378 [162–701.5] *	272 [126–550] *	365.5 [158.5–658] *	0.444	625 [281–1150] *	**<0.001**
Transfusion	27 (11.5%)	15 (13.8%)	6 (8.0%)	0.227	6 (12.0%)	0.944
CD classification ≥ 3a	50 (21.3%)	24 (22.0%)	9 (12.0%)	0.082	17 (37.0%)	**0.021**
Mortality	2 (0.9%)	2 (1.8%)	0	0.514	0	0.999
Postoperative hospital stays	21 [13–35] *	21 [13–34.5]*	18 [13–27] *	0.307	28 [18–53] *	**0.003**
Adjuvant treatment	177 (75.3%)	86 (78.9%)	57 (76.0%)	0.642	34 (66.7%)	0.105
pT1/pT2/pT3/pT4	24/7/202/2	16/4/88/1	8/1/66/0	0.492	0/2/48/1	**0.040**
pN0/pN1	103/132	53/56	28/47	0.130	22/29	0.910
pM0/pM1	228/7	106/3	71/4	0.446	51/0	0.352
R0/R1/R2	197/34/4	87/21/1	66/6/3	0.145 †	44/7/0	0.554 ^†^
pStage I/II/III/IV	26/198/3/8	17/88/1/3	5/64/1/5	0.194	4/46/1/0	0.342
Tumor size (mm)	27 [19–40] *	25 [17–33.5] *	29.5 [20–40] *	**0.040**	32 [25–55] *	**<0.001**
No. of harvested LNs (*n* = 163)	25 [14–37] *	27 [14.5–43.5] *	20.5 [12.25–32.5] *	**0.022**	33 [18.5–41] *	0.074
pA positive	33 (13.9%)	13 (11.9%)	13 (17.3%)	0.301	7 (13.7%)	0.941
pPV positive	99 (41.9%)	34 (31.2%)	37 (49.3%)	**0.015**	28 (54.9%)	**0.040**
pOO positive	35 (14.8%)	5 (4.6%)	20 (26.7%)	**<0.001**	10 (19.6%)	0.285

The number of patients (percentage). * Median [interquartile range]. ^†^ Compared with R0 vs. R1 + R2. The significant values are indicated in bold. Pb, pancreatic body (and/or neck); Pbt, pancreatic body and tail; Pt, pancreatic tail; BMI, body mass index; HU, Hokkaido University; SMU, Sapporo Medical University; KJ, Teine-Keijinkai Hospital, Kin; Kin-ikyo Chuo Hospital, DP-CAR; distal pancreatectomy with celiac axis resection; DP: distal pancreatectomy; SPDP: spleen-preserving distal pancreatectomy, CD classification; Clavien–Dindo classification, pT; pathological tumor category, pN; pathological node category, pM; pathological metastasis category, R; pathological residual tumor factor, No. of harvested LNs, number of harvested lymph nodes, pA, pathological arterial system invasion, pPV, pathological portal venous system invasion, pOO, pathological invasion of other organs.

**Table 2 cancers-14-04409-t002:** Distribution of lymph node metastasis based on tumor site and efficacy index according to the 5-year survival rate.

	Total Patients *n* = 235	Pb *n* = 109			Pbt *n* = 51			Pt *n* = 75		
	Frequency of Metastasis	Frequency of Metastasis	5-Year OS Rate (%)	5-Year OS Index	Frequency of Metastasis	5-Year OS Rate (%)	5-Year OS Index	Frequency of Metastasis	5-Year OS Rate (%)	5-Year OS Index
#8a/p	18 (7.7%)	12 (11.0%)	21.09	2.32	4 (7.8%)	37.50	2.93	2 (2.7%)	0.00	0.0
#10	16 (6.8%)	0	N.A.	N.A.	4 (7.8%)	0.00	0.0	12 (16.0%)	38.89	6.22
#11d	47 (20.0%)	5 (4.6%)	0.00	0.0	14 (27.5%)	0.00	0.0	28 (37.3%)	4.76	1.77
#11p	62 (26.4%)	39 (35.8%)	28.37	10.16	13 (25.5%)	19.44	4.96	10 (13.3%)	14.81	1.97
#14p	3 (1.3%)	1 (0.9%)	100.00	0.9	1 (2.0%)	0.00	0.0	1 (1.3%)	0.00	0.0
#16b1	5 (2.1%)	2 (1.8%)	0.00	0.0	1 (2.0%)	0.00	0.0	2 (2.7%)	0.00	0.0
#18	37 (15.7%)	17 (15.6%)	10.98	1.71	8 (15.7%)	12.50	1.96	12 (16.0%)	11.54	1.85

Pb, pancreatic body (and/or neck); Pbt, pancreatic body and tail; Pt, pancreatic tail; OS, overall survival; #8a/p, lymph nodes (LNs) along the common hepatic artery; #10, LNs at the splenic hilum; #11d, LNs along the distal splenic artery; #11p, LNs along the proximal splenic artery; #14p, LNs along the proximal superior mesenteric artery; #16b1, LNs around the abdominal aorta; #18, LNs along the inferior margin of the pancreas; N.A., not applicable.

**Table 3 cancers-14-04409-t003:** Univariate and multivariate analyses of the influence of various factors on overall survival.

	Univariate Analysis		Multivariate Analysis	
	HR (95%CI)	*p* Value	HR (95%CI)	*p* Value
Age > 70/ ≤ 70	1.521 (1.090–2.122)	0.014	1.365 (0.962–1.938)	0.082
CA19−9 > 100/ ≤ 100	2.155 (1.539–3.017)	0.001	**1.451 (1.009–2.086)**	**0.045**
pN 1/0	2.253 (1.581–3.211)	0.001	1.485 (0.965–2.284)	0.072
pPV 1/0	1.908 (1.364–2.669)	0.001		
pA 1/0	2.504 (1.625–3.857)	0.001	1.453 (0.911–2.086)	0.117
pOO 1/0	1.674 (1.092–2.568)	0.018		
Tumor location Pbt/Pb or Pt	0.895 (0.609–1.314)	0.571		
R1 + 2/0	1.473 (0.962–2.256)	0.075	1.578 (0.998–2.496)	0.051
Tumor size >20 mm/≤20 mm	2.381 (1.588–3.569)	0.001	**1.648 (1.053–2.579)**	**0.029**
NAT +/−	1.190 (0.745–1.902)	0.466		
AT −/+	1.979 (1.393–2.812)	0.001	**2.191 (1.513–3.172)**	**0.001**
#8a/p LNs +/−	1.254 (0.658–2.391)	0.491		
#10 LNs +/−	1.229 (0.664–2.277)	0.512		
#11d LNs +/−	2.722 (1.874–3.955)	0.001	**1.914 (1.219–3.005)**	**0.005**
#11p LNs +/−	1.415 (0.978–2.048)	0.065		
#14p LNs +/−	2.660 (0.652–10.850)	0.173		
#16b1 LNs +/−	2.027 (0.748–5.492)	0.165		
#18 LNs +/−	1.740 (1.143–2.650)	0.010		

HR: hazard ratio; 95%CI: 95% confidence interval; pN: pathological node category; pPV: pathological portal venous system invasion; pA: pathological arterial system invasion; pOO: pathological invasion of other organs; Pb: pancreatic body (and/or neck); Pbt: pancreatic body and tail; Pt: pancreatic tail; R: pathological residual tumor factor; NAT: neoadjuvant treatment; AT: adjuvant treatment. LNs; lymph nodes. Significant values in multivariate analyses are shown in bold.

## Data Availability

The data presented in this study are available on request from the corresponding author.

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
