# Peer review of "Appropriate Lymph Node Dissection Sites for Cancer in the Body and Tail of the Pancreas: A Multicenter Retrospective Study"

_cancers, 2022, doi:10.3390/cancers14184409_

Round 1

Reviewer 1 Report

The article is of scientific interest and can be published. I only advised to broaden the conclusions. However, given the scientific validity, it can be widely accepted.

Author Response

Responses to the Reviewers' suggestion 

Reviewer #1

Comments:

The article is of scientific interest and can be published. I only advised to broaden the conclusions. However, given the scientific validity, it can be widely accepted.

Our response:

Thank you very much for providing your comments. We are thankful for the time and energy you expended.

Reviewer 2 Report

This is a very interesting study , clinically important.

I have some suggestions .In the Introduction section the Authors are invited to make some mention to possible  patient  immunodepression,in particular after surgery and transfusion. Immunodepression, indeed, can impact on the initiation , continuation of chemotherapy  , doses of anticancer agents.In addition , immunodepression can increase the risks for postsurgery complications including impaired surgical wound healing.

In the Results section, paragraph 3.3(Long-term outcomes) variables such as serum albumin, Haemoglobin levels , blood lymphocytes both in absolute values and % of Total white cells, blood neutrophil/ lymphocyte ratio should enter ,if the case, the multivariate analysis.All these factors might influence long- term outcomes and are easily available. Given the nature of the investigation( retrospective , multicenter study) , if it is difficult to get these variables , the Authors are invited to make some mention in Limits paragraph.

Author Response

Responses to the Reviewer 2 suggestion 

Reviewer #2

Point 1: I have some suggestions. In the Introduction section the Authors are invited to make some mention to possible  patient  immunodepression,in particular after surgery and transfusion. Immunodepression, indeed, can impact on the initiation , continuation of chemotherapy  , doses of anticancer agents.In addition , immunodepression can increase the risks for postsurgery complications including impaired surgical wound healing.

Our response 1: Thank you very much for providing important comments. We are thankful for the time and energy you expended. 
As you advised, we have added the sentences in the introduction part regarding the association between patient immunodepression and complication. (P.2 L.5-12)

“ In general, major surgery induced tissue damage followed by an extensive immuno-suppressive response. Intraoperative blood transfusion further augments the predomi-nately immunosuppressive response seen secondary to tissue damage [4]. Perioperative blood transfusion following pancreatectomy for PDAC is associated with poor short- and long-term outcomes [5]. Immune suppression following surgery has been implicated in increased short-term postoperative complications, especially infectious complications [4]. Therefore, treatment of PDAC may require surgery that does not induce a postoperative immunosuppressive response. ”

Point 2: In the Results section, paragraph 3.3(Long-term outcomes) variables such as serum albumin, Haemoglobin levels , blood lymphocytes both in absolute values and % of Total white cells, blood neutrophil/ lymphocyte ratio should enter ,if the case, the multivariate analysis.All these factors might influence long- term outcomes and are easily available. Given the nature of the investigation( retrospective , multicenter study) , if it is difficult to get these variables , the Authors are invited to make some mention in Limits paragraph.

Our response 2: Thank you so much for a valuable comment. It was difficult to collect the laboratory data on immune response and nutritional status such as preoperative leukocyte fractionation and serum albumin from all the patients. We have therefore described the reasons for this in the limitation paragraph. (P.10 L10-13)

“The laboratory data representative of immune response and nutritional status should be included in the multivariate analysis of long-term outcomes, but the wide variation among centers and time periods made analysis difficult for the multicenter retrospective study. ”

Round 2

Reviewer 2 Report

The Authors of the article  have answered appropriately and correctly.Therefore the manuscript warrants publication. Best regards